



# Robust Bilinear Rotations II

Yannik T. Woordes[1] and Burkhard Luy[1,*]

[1] Institute of Organic Chemistry and, Institute for Biological Interfaces 4 - Magnetic Resonance , Karlsruhe Institute of Technology (KIT) , Hermann-von-Helmholtz-Platz 1 , 76344 Eggenstein-Leopoldshafen , Germany , Tel.: +49 721 608 29353 (Y.T.W.), email: yannik.woordes@kit.edu, ORCID-ID: 0009-0007-5550-6788 , Tel.: +49 721 608 29360 (B.L.), email: burkhard.luy@kit.edu , ORCID-ID: 0000-0001-9580-6397

**Correspondence:** Burkhard Luy (burkhard.luy@kit.edu)

**Abstract.** Bilinear rotations are essential building blocks in modern NMR spectroscopy. They allow the rotation of an isolated spin without couplings, i.e. bilinear intereactions, in one way, while rotating spins with a matched coupling in another way. Different classes of rotations form the different bilinear rotations with acronyms BIRD, TANGO, BANGO, and BIG-BIRD. All original elements have in common that hard pulses limit bandwidths and that defined rotations for coupled spins are only possible for a narrow range of coupling constants. We recently introduced the COB-BIRD with a general optimization procedure to obtain robust bilinear rotations well-compensated with respect to couplings, offsets, and $B_1$-inhomogeneities (Y. T. Woordes et al., Sci. Adv. 11 (2025), eadx7094). Here we show a fundamental principle on how the COB-BIRD can be used to construct all types of bilinear rotations with the same improved robustness covering a coupling range of 120-250 Hz. In addition, a construction principle for universal rotation pulses is adapted to produce bilinear rotations from INEPT-type transfer elements, allowing the construction of bilinear rotations also for higher coupling ranges from e.g. COB3-INEPT with coupling compensation in the range of 120-750 Hz. After introducing the two fundamental design principles, example sequences of the four classes of bilinear rotations and different degrees of robustness are derived and characterized in theory and experiment. In addition, a highly useful HMBC/ASAP-HSQC-IPE-COSY supersequence is introduced with a (COB-)BANGO element for Ernst-angle type excitation. Finally, BIRD-decoupled $J$-resolved INEPT experiments with extreme compensation for partially aligned samples with with total couplings ranging from 47 Hz up to 434 Hz are demonstrated.

## 1 Introduction

Bilinear rotation elements represent fundamental building blocks in NMR spectroscopy with a large variety of applications. The success story started with bilinear rotation decoupling (BIRD) by Garbow, Weitekamp, and Pines in 1982 (Garbow et al., 1982) with a spin system selective 180° rotation as the central element. The basic element has a decade later been generalized with a systematic nomenclature by Uhrín, Liptaj, and Kövèr (Uhrín et al., 1993), and since then has been used for spectral cleanup (Kurz et al., 1991; Schmieder et al., 1991), homonuclear decoupling (Krishnamurthy and Casida, 1988; Furrer et al., 2007; Lupulescu et al., 2012; Sakhaii et al., 2009; Kiraly et al., 2015; Aguilar et al., 2011; Donovan and Frydman, 2015; Haller et al., 2022; Gyöngyösi et al., 2021; Saurí et al., 2015, 2017; Kaltschnee et al., 2014), enhanced measurement of couplings (Fehér et al., 2003; Reinsperger and Luy, 2014; Schulze-Sünninghausen et al., 2017; Timári et al., 2014, 2016),





enhanced resolution in a $J$-evolved dimension (Furrer et al., 2007), and many more applications. Next to BIRD variants, several other bilinear rotations exist like TANGO (Wimperis and Freeman, 1984) and BANGO (Sørensen, 1994) universal coupling-dependent rotations, or corresponding BIG-BIRD (Sørensen, 1994) and TIG-BIRD (Briand and Sørensen, 1998) elements to transfer initial polarization into any desired magnetization depending on the absence or presence of a large one-bond heteronuclear coupling. The basic BIRD element has been extended for better robustness, leading to G-BIRD (Emetarom et al., 1995; Mackin and Shaka, 1996) with enhanced cleanup, CAGE-BIRD (Koskela et al., 2004) for better homonuclear coupling handling, or BASEREX (Haller et al., 2019, ?; Sebák et al., 2022) for selective treatment of isotope-labeled samples. Nevertheless, all basic bilinear rotations have in common that they are relatively sensitive to coupling mismatch and offset effects (Garbow et al., 1982; Lupulescu et al., 2012; Torres et al., 1990; Bigler et al., 2024; Woordes et al., 2025). The issue has been recognized early on and $J$- as well as offset-compensated BIRD elements have already been proposed in the seminal BIRD paper (Garbow et al., 1982), but only a recent systematic study of correspondingly coupling-, offset-, and $B_1$-compensated BIRD elements (COB-BIRD) has provided a robust element for isotropic and very weakly coupled partially aligned samples (Woordes et al., 2025).

In the light of novel applications like fast-pulsing and interleaved acquisition sequences (Yong et al., 2021; Hansen et al., 2021; ?; Nagy et al., 2021; Timári et al., 2022; Schulze-Sünninghausen et al., 2025), robust sequences are also needed for bilinear rotation elements beyond BIRD. In the following manuscript, we therefore derive a general scheme that reduces all bilinear rotations to its central spin system-selective refocusing element, with which the robustness of the COB-BIRD can be transferred to all other bilinear rotations, i.e. robust COB-TANGO, COB-BANGO, COB-BIG-BIRD, COB-BASEREX. In addition, we extend and apply a previously derived symmetry scheme (Luy et al., 2005) to deduce sequences with $J$-compensation over even larger ranges of couplings than COB-BIRD by using already existing compensated INEPT-type transfer elements (Ehni and Luy, 2012, 2014). All derived elements are studied in both theory and experiment. We also present two applications demonstrating the usefulness of the extended $J$-coupling range and the benefit of a robust COB-BANGO element in modern NORD-type, fast pulsing supersequence schemes.

## 2 Generalizing COB-enhanced Bilinear Rotations

Bilinear rotations are spin system selective heteronuclear building blocks that distinguish spins $I$, that are not directly coupled to a heteronucleus, from spin systems $IS$, where the spin $I$ is coupled to a spin $S$ via a large heteronuclear coupling $J$. In BIRD elements, the difference between uncoupled and coupled spins usually lies in the phase of a transverse $\pi$-rotation. TANGO elements, instead, provide a 90° (or any arbitrary $\beta$-) pulse for one and either 0° or 180° for the other type of spin system. BANGO elements allow universal rotations with arbitrary flip angles $\beta^I$ and $\beta^{IS}$ for the spin systems. While all these elements allow rotations only about a specific rotation axis, BIG-BIRD rotates initial $I_z$ polarization into any final position that can be reached by effective $\beta^I_{\phi^I}$ and $\beta^{IS}_{\phi^{IS}}$ rotations, introducing the effective phases $\phi^I$ and $\phi^{IS}$ for the two spin systems. As it turns out, on the one hand, the different types of bilinear rotations manipulate spins in very different ways, but, on the other





hand, they all have an identical central building block in common that is responsible for their spin system-selective properties (cf. Fig. 1).

In Fig. 1 A the general scheme of all mentioned bilinear rotation elements is shown. While flanking pulses make up the
difference in the various bilinear rotations, the central refocused delay of overall duration $1/J$ is common to all of them. It is in all cases this central element that is responsible for the distinction of $I$ and $IS$ spin systems. With central $180^\circ_x$ pulses on both nuclei, the refocused delays provide a $\pi_x$-rotation for the uncoupled $I$ spin and a $\pi_y$-rotation, if the $IS$ spin system is coupled with the matched heteronuclear coupling $J$. It is thereby important to note, that the rotation transforms all three Cartesian components in a defined way.

Understanding this common design principle of all bilinear rotations, it is sufficient to make the central refocused delay robust in the desired way to significantly enhance all different elements at the same time. As such, the central blocks derived in the COB-BIRD (Woordes et al., 2025) can directly be used to make any type of basic bilinear rotation robust with respect to a $J$-coupling range of $120-250$ Hz and an offset range determined by the used shaped pulses, i.e. 37.5 kHz for the previously reported $J$-compensated BUBI (Ehni and Luy, 2013) and BUBU (Ehni et al., 2022) pulse sandwiches, respectively. As the
flanking pulses of a conventional BIRD element are 90° pulses, corresponding pulses need to be eliminated from the COB-BIRD element. We then obtain a robust universal rotation type element that equally rotates all remote protons by 180° around $x$ and all directly bound protons by 180° around $y$. The resulting COB central element can be used for the construction of BIRD, TANGO, BANGO, and BIG-BIRD elements (Fig. 1 B,D,F) as shown in Fig. 1 C,E,G. Together with the sequences, also the corresponding offset vs. coupling profile of the general central element of the conventional bilinear rotations (Fig. 1 H) as
well as the central COB element (Fig. 1 I) are given.

## 3 Construction Principle of COB-enhanced Bilinear Rotations

Having derived the central necessary condition to make bilinear rotations more robust, we can also use it to construct highly compensated central $\pi$-rotation elements from already existing robust INEPT-type transfer elements. Considering just the initial pulse and the coupling evolution of such elements, they can be considered a $J$-selective transfer element leading to $I_z \rightarrow I_x$ for
a spin without heteronuclear coupling and to $I_z \rightarrow 2I_yS_z$ for a matching $J = 1/2\Delta$-coupling constant. This behavior can be considered a point-to-point transformation with an effective flip angle of $(\pi/2)_y$ for a spin without coupling and an effective flip angle of $(\pi/2)_{-x}$ for the coupled case with respect to the coupling Hamiltonian $\mathcal{H}_J = 2\pi JI_zS_z$. As long as only $\pi$ pulses are applied and no mixing of spin states is apparent on the $S$ spin, this transformation can be considered homomorph to a point-to-point pulse transforming $I_z \rightarrow I_y$ (see Fig. 2 A).

The target for the BIRD bilinear rotation, on the other hand, are overall $\pi$ rotations represented by the propagator $U_T = \exp(-i\pi 2I_{\gamma_I}S_{\gamma_S})$ with $\{\gamma_I, \gamma_S\} = \{x, x\}$ for two uncoupled spins $I$ and $S$, and $\{y, y\}$ for the coupled $IS$ spin system. Again, this can be translated to a homomorphous frequency-selective universal rotation pulse with target propagators $U_T(0) = \exp(-i\pi I_x)$ and $U_T(J) = \exp(-i\pi I_y)$, as long as we are only interested in the effective $I$ spin rotations. As it turns out, a construction principle exists, for which such $\pi$-pulses can be created from corresponding $\pi/2$ pulses: for an effective universal





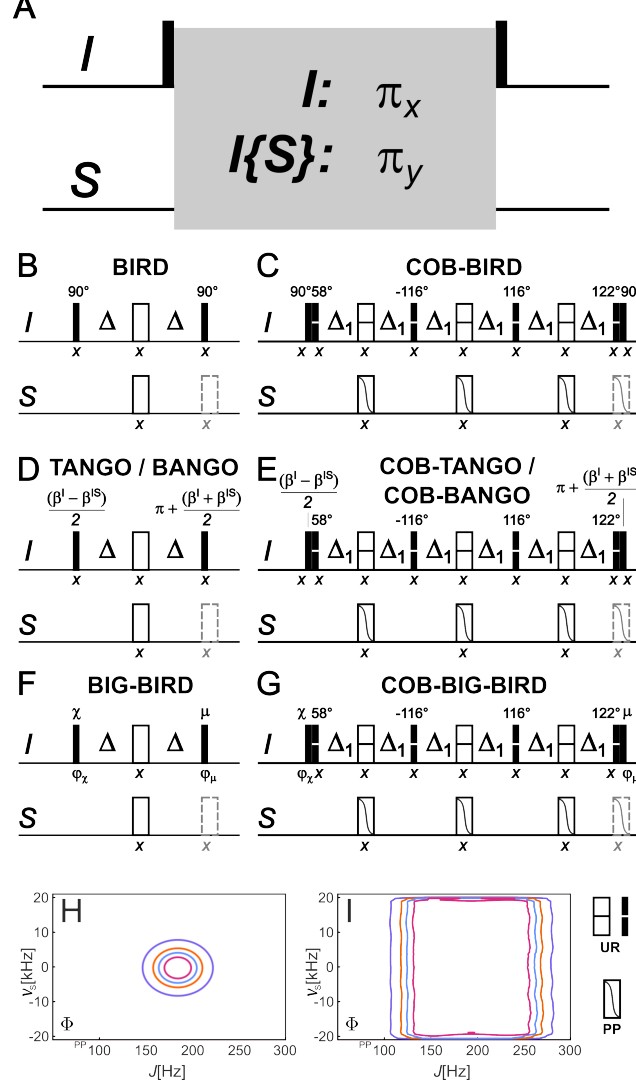

**Figure 1.** Construction of any type of bilinear rotation from a single spin system-selective $\pi$ rotation element. The underlying insight into the construction scheme is the fact that any of the basic bilinear rotations in literature is based on two flanking pulses around the very same rotation scheme that provides a $\pi_x$ rotation for an uncoupled spin $I$ and a $\pi_y$ rotation for an $IS$ spin system with a coupling constant that is matched to sequence (A, rotation element represented by the gray box). Corresponding BIRD, TANGO, BANGO, and BIG-BIRD bilinear rotations are obtained by specifically chosen flanking pulses (B,D,F). If the simple refocused delay with $\Delta = 1/(2\,J)$ is replaced by a more robust spin system-selective $\pi$ rotation element, like the one here derived from the recently published COB-BIRD bilinear rotation (Woordes et al., 2025), any type of bilinear rotation can be obtained by simply using corresponding flanking pulses with the element (C,E,G). Be aware that consecutive pulses before and after the rotation element are represented for clarity as separate pulses, but may be combined to a single pulse (or universal rotation pulse shape) in a particular application. Please also note the convention for point-to-point (PP) and universal rotation (UR) shaped pulses in the lower right corner taken from reference (Ehni and Luy, 2014). Corresponding robustness with respect to the coupling $J$ and the offset of the heteronucleus $\nu_S$ for the two rotation elements is simulated using the quality factor $\Phi$. $\Delta_1$ is set to 2.583 ms (Woordes et al., 2025).





**Figure 2.** Construction principle for obtaining a bilinear rotation element from an INEPT-type transfer element. The principle uses the equivalence of a heteronuclear coupled two spin system (A) to an offset selective shaped pulse (A') as both interaction describe a rotation in a 3D Hilbert (sub-)space. The previously reported construction principle from an effective point-to-point $\pi/2$ pulse to a universal $\pi$ rotation (Luy et al., 2005) (B) can then be extended to construct a BIRD bilinear rotation from a reduced INEPT sequence following essentially the same construction procedure (B'). A simple selective excitation scheme (C, top) equivalent to an on-resonant, reduced INEPT element (C', top) will then be constructed to give a selective $\pi$ rotation (D, top) and the corresponding bilinear rotation (D', top), respectively. Introducing a pair of $180°$ pulses (D', drawn in gray) results in the offset-compensated classical scheme. Corresponding offset and $J$ dependencies of excited coherences and magnetization transfers are given below the sequences. Note that the factor 2 in the offset vs. $J$ dependence results from the rotation frequency of the two interactions. Using a more elaborate INEPT-type transfer element like the reduced COB3-INEPT (Ehni and Luy, 2014) (E, E'), highly $J$-compensated bilinear rotations can be constructed (F, F'). Note that $30°$ pulses (drawn in gray) are added to the original COB3-INEPT sequence to ensure the transfer $I_z \rightarrow I_x$ for vanishing offsets or $J$ couplings, respectively (E, E', F, F'). Delays are taken from (Ehni and Luy, 2014) to be $\Delta_2 = 4.221$ ms, $\Delta_3 = 2.0634$ ms, $\Delta_4 = 2.14$ ms, and $\Delta_5 = 1.075$ ms.





rotation pulse with flip angle $2\beta$, a time- and phase-reversed and an original point-to-point pulse with effective flip angle $\beta$ starting from $I_y$ need to be applied consecutively to obtain the desired rotation (Luy et al., 2005). In the case of a point-to-point pulse transferring $I_z$ to $I_y$ at a defined offset $\nu$, this can be reformulated to apply first the phase-reversed point-to-point pulse, followed by the time-reversed point-to-point pulse to construct a universal rotation $\pi_x$ pulse at offset $\nu$ (Fig. 2 B left). Using the homomorphous relation to the INEPT-type transfer element and heteronuclear coupling evolution, a corresponding bilinear $\pi$ rotation element can be constructed (Fig. 2 B right).

Using the homomorph of the first pulse and the delay of a conventional INEPT-element, which leads to $\pi/2$ rotations with offset-dependent phase (Fig. 2 C left), this can be used to construct a frequency-selective universal $\pi$ rotation element (Fig. 2 D left). This element directly translates into the coupling evolution case, for which the original INEPT-derived transfer element directly creates a conventional BIRD element. All it needs is the insertion of two pairs of 180° pulses to give the BIRD-element the needed robustness with respect to chemical shift offsets (Fig. 2 D right). In Fig. 2 E,F the very same exercise is accomplished using the highly $J$-compensated COB3-INEPT sequence previously published in (Ehni and Luy, 2014). The original COB3-INEPT was optimized only for the transfer $I_z \rightarrow 2I_yS_z$ for a given heteronuclear coupling constant $J$ and the condition 120 Hz $\leq J \leq$ 750 Hz. The transfer $I_z \rightarrow I_x$ for the case $J = 0$ Hz, however, was not taken into account. This is easily corrected by an additional $30°_x$-pulse right after excitation, which compensates the sum of all other pulses applied on-resonant and without coupling being present. The overall COB3-BIRD bilinear rotation sequence looks complex, but a very high $J$-compensation is achieved. As many pulses are applied during the sequence, it should be noted that only the use of well-compensated shaped pulses (provided in (Ehni and Luy, 2014)) will lead to the desired result in experiments for a given range of chemical shift offsets.

We applied the construction principle also to the original COB-INEPT (Ehni and Luy, 2012) and the second broadband $J$-compensated COB3-INEPT sequence given in (Ehni and Luy, 2014), resulting in the sequences that are referred to by COB-BIRDcp and COB3-BIRDcp in the experimental verification section.

## 4 Performance with respect to $J$

To verify the validity of the various COB-enhanced bilinear rotation elements of Fig. 1, we recorded $J$-dependency profiles for four example bilinear rotations (BIRD, TANGO, BANGO, BIG-BIRD) for different central inversion elements (Fig. 3).

While we varied the heteronuclear $J$-coupling for simulations, experimental incrementation of a coupling constant is not feasible and we used varying delays instead to effectively measure the coupling dependence: as the evolution during all delays $\Delta_i$ is given by $\cos \pi J \Delta_i$, an increasing coupling constant $J$ can be also emulated by scaling all delays $\Delta_i$ with a common factor. In addition, we compensated potential signal losses due to increasing delays by a compensation delay after the bilinear rotation element, ensuring approximately identical transverse relaxation periods for all effective $J$-couplings (Fig. 3 C). For the acquisition of $J$-profiles we used a 1:2 mixture of unlabeled acetate and $^{13}C_2$-acetate with a measured coupling constant of $^1J_{CH} = 125$ Hz. The ratio of unlabeled vs. labeled acetate results in three lines of approximately equal intensities in an uncoupled $^1H$ spectrum.

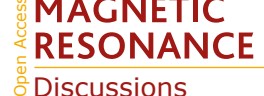

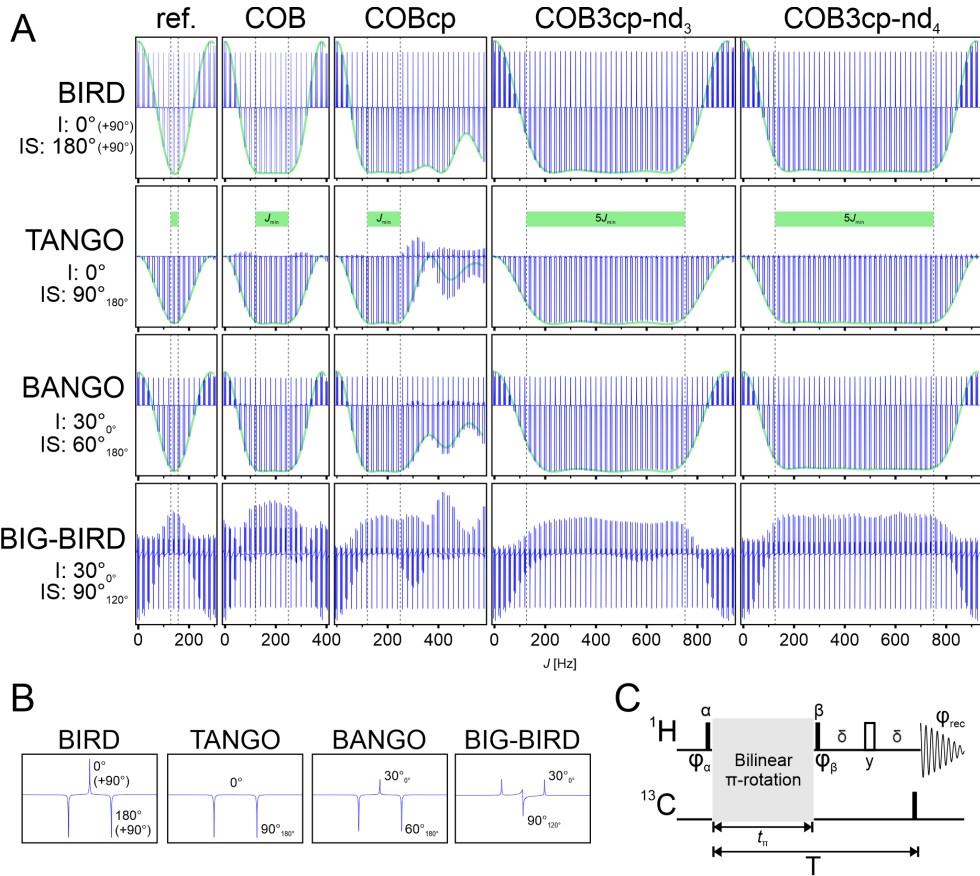

**Figure 3.** $J$-dependencies of various example bilinear rotations. Experimental verification of the coupling profiles of the various bilinear $\pi$-rotation elements by their application in a BIRD, TANGO, BANGO and BIG-BIRD pulse element. (A) Simulated coupling profile (green line) superimposed over the experimentally obtained data (blue spectral lines). Experimental data has been obtained from 120 mM sample of sodium acetate (including 66% sodium $^{13}C_2$-acetate) in 4:1 DMSO-$d_6$ : $D_2O$. From left to right the spectra show the original sequences with refocused delays (ref. – delays matched to 145 Hz $J$-coupling), the COB-BIRD (COB), the COB-BIRDcp (COBcp), the COB3-BIRDcp-nd3 (COB3cp-nd3), and the COB3-BIRDcp-nd4 (COB3cp-nd4) as the central spin system selective $\pi$ rotations and from top to bottom the respective elements are applied as a BIRD, TANGO, BANGO and BIG-BIRD with their respective flip angles for the I and IS spin systems shown. As the signal intensity of the BIG-BIRD is modulated by the phase, no simulation is superimposed. Within the TANGO series – using a green bar and dashed black lines – optimized regions of COB-sequences are indicated. For displaying purposes, the last 90° pulse of the BIRD element has been left out as this would result in +/- z magnetization. As the pulses are equal irrespective of the coupling, leaving it out does not affect the results – as can be seen from the correspondence between simulation and experimental data. (B) As an example for each pulse element, the matched case of the ref. spectrum is shown. The central line and outer signals correspond to the $^{12}C_2$-acetate (I) and $^{13}C_2$-acetate (IS), respectively. Respective flip angles for the pulse elements are indicated. (C) Relaxation-compensated pulse sequence used to record the $J$-profiles. Pulses $\alpha$, $\beta$ with phases $\phi_\alpha$, $\phi_\beta$ are adapted to the different types of bilinear rotations, resulting in $(90^\circ_x, 90^\circ_{-x})$ for the BIRD$^d$, $(45^\circ_x, 45^\circ_x)$ for the TANGO, $(60^\circ_x, 30^\circ_x)$ for the BANGO, and $(52.24^\circ_{129.23^\circ}, 135^\circ_{270^\circ})$ for the BIG-BIRD element. Variation in $J$ is emulated by scaling the delays in the $\pi$ rotation elements as described in (Ehni and Luy, 2012, 2014), resulting in scaled overall $\pi$ rotation times $t_\pi$. Relaxation compensation is achieved by adjusting the delay $\delta$ in such a way that the overall duration T stays constant.





The resulting experimental $J$-profiles are given in Fig. 3 A, with each row representing a different bilinear rotation element and each column representing a central $\pi$ rotation element. We chose a BIRD$^{d,X}$ (resulting in an initial $90^\circ_x$ pulse, while the second pulse is canceled with an excitation pulse for the otherwise undetectable polarization), a TANGO($90^{IS}$,$0^I$) (with two $45^\circ_x$ flanking pulses ), a BANGO with $\beta^I = 30^\circ$ and $\beta^{IS} = 60^\circ$ (using a $60^\circ_x$ pulse before and a $30^\circ_x$ pulse after the central element), and a BIG-BIRD sequence with $\beta^I = 30^\circ$, $\varphi^I = 0^\circ$, $\beta^{IS} = 90^\circ$, and $\varphi^{IS} = 120^\circ$ (resulting in $\chi_{\varphi_\chi} = 52.24^\circ{}_{129.23^\circ}$ and $\mu_{\varphi_\mu} = 135^\circ{}_{270^\circ}$. In all cases but the BIG-BIRD profiles we overlaid simulated $J$-profiles as solid red lines over the experimental data for inner (uncoupled) and outer (coupled) multiplet components.

The central $\pi$ rotation elements used for the different $J$-profiles are summarized in Fig. 4 D-H. The original spin system selective $\pi$ rotation is based on a refocused delay of duration $1/J$. The $\pi$ rotation of the recently reported COB-BIRD (Woordes et al., 2025) as derived in Fig. 1 constitutes the second element. Further $\pi$ rotations use the construction principle with previously derived COB-INEPT (Ehni and Luy, 2012) and COB3-INEPT (Ehni and Luy, 2014) building blocks as described in the previous section and Fig. 2. Resulting elements are called COBcp, COB3cp-nd$_3$, and COB3cp-nd$_4$, where cp indicates the use of the construction principle and nd$_i$ the number of delays of the original INEPT-type building block.

The resulting $J$-profiles impressively show the improved robustness with $J$ for all COB-type elements. While the COB $\pi$ rotation has excellent performance over the full coupling range of 120-250 Hz, all other sequences have similar performance over their entire coupling range with few negligible dents in performance. The COB-BIRDcp performs well even over a range significantly exceeding the optimized coupling range, which is not the case for the other bilinear rotations constructed from the COBcp element. The COB3cp building blocks, on the other hand, perform incredibly well over the entire optimized coupling range of 120-750 Hz. Just for COB3cp-nd$_3$ it should be noticed that performance in the range of 120-190 Hz is somewhat compromised.

## 5 Application in NORD-type supersequences

An area in which bilinear rotations elements become increasingly important are so-called fast-pulsing supersequences. Here, bilinear rotation elements other than BIRD play an important role in spin system selective excitation and polarization storage. The development started with the NORD (NO Relaxation Delay) sequence (Nagy et al., 2021) and has found since then several extensions (Timári et al., 2022; Bence Farkas et al., 2022) based on a generalized Ernst-angle scheme (Koos and Luy, 2019; Sørensen, 2024). A very useful NORD supersequence involves an HMBC with a subsequent H2OBC (Nagy et al., 2021), which results in two separate spectra where one shows typical long-range HMBC correlations, while the second spectrum gives almost exclusively one-bond and two-bond $^1H$,$^{13}C$-correlations. For a fast acquisition of the two spectra it is mandatory that only protons with long-range $^1H$,$^{13}C$-couplings are excited for the HMBC to retain the polarization of directly $^{13}C$-bound protons for the subsequent H2OBC. This exactly matches the TANGO/BANGO profile, or, if also specific phase settings are required, the BIG-BIRD element. We implemented the original pulse sequence from (Nagy et al., 2021), but experienced several problems resulting in non-absorptive lineshapes in the H2OBC sub-spectrum. We therefore modified the sequence to not use the H2OBC, but an ASAP-HSQC-IPE-COSY with the very same information content instead. This novel sequence





is essentially an HSQC-COSY, which uses a CLIP-COSY (**?**) for coherence transfer as in Ref. (Gyöngyösi et al., 2021), but also retains unused polarization during $^1H,^1H$-coherence transfer periods. As a consequence, the perfect echo element (**?**) is extended to an isotropic perfect echo (Haller, 2021), which will be described in more detail in a separate publication. The ASAP-HSQC, on the other hand, is an established fast pulsing sequence using the generalized Ernst angle and maintaining unused polarization (Schulze-Sünninghausen et al., 2014; Becker and Luy, 2015; Schulze-Sünninghausen et al., 2017; Becker et al., 2019; Schulze-Sünninghausen et al., 2025). The sequence with all experimental details is given in Fig. 4 A.

In the conventional NORD approach a BIG-BIRD bilinear rotation is used for spin system selective excitation, which, of course, would be good to be replaceed by a more robust element. The BIG-BIRD sequence in the NORD-HMBC-H2OBC experiment is needed to compensate phase-twists in the H2OBC sub-spectrum. The ASAP-HSQC-IPE-COSY, instead, does not need phase adjustment and a BANGO element can be used. For obtaining spectra shown in Fig. 5, we used 60° BANGO and COB-BANGO elements for comparison on a test sample containing several compounds with triple bonds. It must first be stated, that the HMBC-ASAP-HSQC-IPE-COSY supersequence leads to HMBC and HSQC-COSY-type subspectra of very high quality.

Irrespective of the spin system selective excitation element, the Ernst angle type excitation as well as the application of the ASAP-HSQC sequence lead to very fast pulsing times without compromise in spectral quality and even improved sensitivity. In the direct comparison of BANGO and COB-BANGO experiments, the HMBC (Fig. 5) and the ASAP-HSQC-IPE-COSY (Fig. 6) need to be compared separately.

The HMBC sub-spectrum in both cases is essentially identical. This behavior is expected, since the HMBC only uses polarization of $I$ spins without direct coupling and the excitation only relies on the performance of corresponding $^1$H pulses, which were all well-compensated optimal control derived shaped pulses. The situation is different for the ASAP-HSQC-IPE-COSY subspectrum, for which the bilinear rotation elements need fitting coupling constants. To achieve best possible performance already for the BANGO sequence, especially optimized universal rotation 30° and 150° shaped pulses together with the BUBI refocusing and inversion pulse sandwich (Ehni and Luy, 2013) have been applied in combination with an overall delay matched to $J = 185$ Hz. As such, only slight improvements are expected for the COB-BANGO sequence for particularly mismatched coupling constants. Indeed, aromatic signals with approximately matching couplings do not show improvements worth mentioning and also signals involed in triple bonds with a coupling of ≈240 Hz result in signal intensities improved only by about 5%. Methyl groups, instead, show improvements exceeding 30% signal intensity. We would have expected similar improvements for $CH_2$ groups, but the signal with a multitude of homonuclear couplings and particularly the large $^2J_{HH}$ coupling shows only a small enhancement on the order of 6%. Still, the COB-BANGO in all cases shows better performance than the conventional BANGO with well-compensated shaped pulses.

## 6 Application invovling partially aligned samples

While conventional isotropic NMR samples very rarely show coupling constants outside the 120-250 Hz range, partially aligned samples with residual dipolar couplings on top of the scalar couplings can lead to total couplings easily exceeding





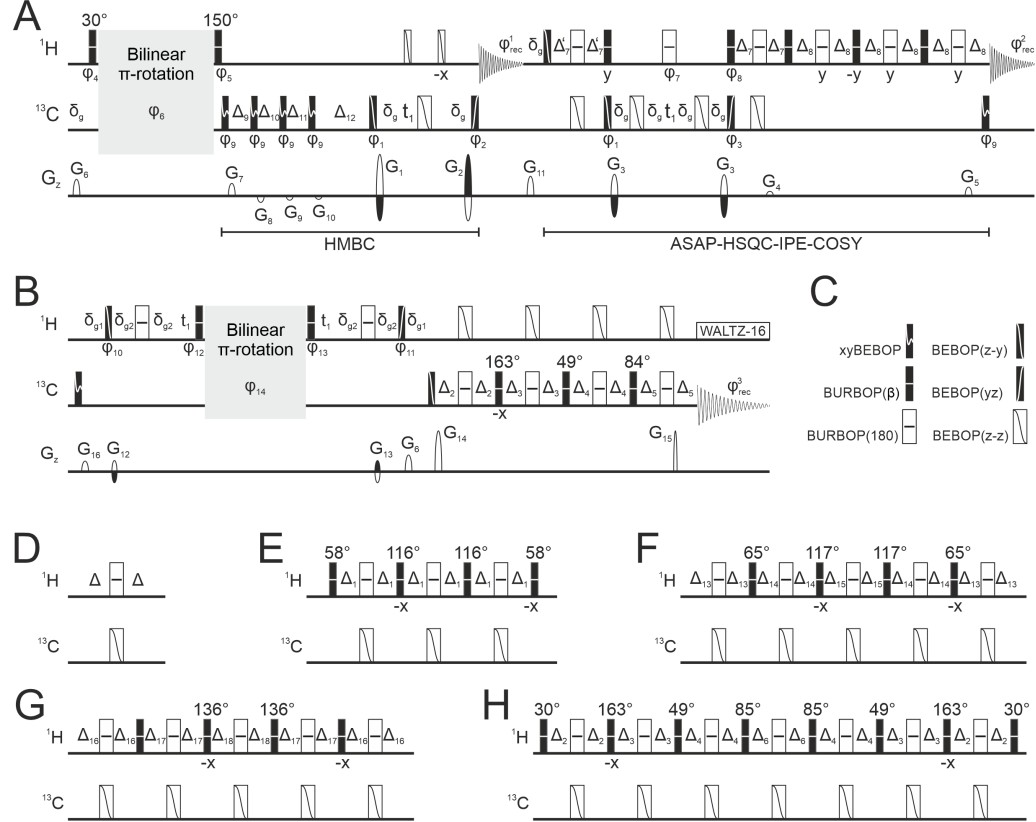

**Figure 4.** Detailed representation of the pulse sequences used in this article. (A, B) The full pulse sequences of the NORD supersequence (A) and the $\{^1\text{H}\}, ^{13}\text{C}$-$J$-INEPT (B) where the gray box is replaced by the respective bilinear $\pi$-rotation elements (D-H). The individual bilinear rotations are (D) BIRD, (E) COB-BIRD, (F) COB-BIRDcp, (G) COB3-BIRDcp-nd$_3$, and (H) COB3-BIRDcp-nd$_4$ (nomenclature explained in the main text). All pulses are applied as shaped pulses with their respective identification presented in (C), if no flip angle is annotated above the BURBOP($\beta$) type pulses, a 90° flip angle is meant. All phases are along x, unless indicated otherwise. Phases are cycled following $\phi_1 = $ x, -x, $\phi_2 = $ x, -x, -x, x, $\phi_3 = $ x, x, -x, -x, -x, -x, x, x, $\phi_4 = $ x, $\phi_5 = $ 4(x), 4(-x), $\phi_6 = $ 4(y), 4(x), $\phi_7 = $ x, y, x, y, y, x, y, x, $\phi_8 = $ x, -x, x, -x, -x, x, -x, x $\phi_9 = $ x, y, -x, -y, x, -y, -x, y, $\phi_{10} = $ 4(x), 4(-x), $\phi_{11} = $ 8(x), 8(-x), $\phi_{12} = $ y, -y, $\phi_{13} = $ -y, y, $\phi_{14} = $ x, x, -x, -x, $\phi^1_{\text{rec}} = $ x, x, -x, -x, -x, -x, x, x, $\phi^2_{\text{rec}} = $ x, x, -x, -x, and $\phi^3_{\text{rec}} = $ 4(x), 8(-x), 4(x), where the phase programs of $\phi_6$ and $\phi_{14}$ are applied on all proton pulses of the bilinear $\pi$-rotation on top of the relative phases within the element. The delays correspond to $\Delta_1 = 2.583$ ms, $\Delta_2 = 2.1105$ ms, $\Delta_3 = 1.0317$ ms, $\Delta_4 = 1.07$ ms, $\Delta_5 = 0.5375$ ms, $\Delta_6 = 2 * \Delta_5 = 1.075$ ms, $\Delta_7 = 1/(4 * {}^1J_{\text{CH}})$, $\Delta_8 = > 1/(4 * J_{\text{HH}}) \approx 30$ ms recommended, $\Delta_9$, $\Delta_{10}$, and $\Delta_{11}$ are the three-fold low pass $J$-filter delays, $\Delta_{12} \approx 1/(2 * {}^3J_{\text{CH}})$ which was matched to an 8 Hz coupling, $\Delta_{13} = 0.394$ ms, $\Delta_{14} = 2.134$ ms, $\Delta_{15} = 2.983$ ms, $\Delta_{16} = 0.5401$ ms, $\Delta_{17} = 1.065$ ms, and $\Delta_{18} = 2.1404$ ms. In the ASAP-HSQC generalized Ernst-angle excitation is achieved by scaling $\Delta'_7 = (\beta/90°)\Delta_7$ for an effective $\beta$-flip angle (Schulze-Sünninghausen et al., 2014, 2017). The gradients used in the sequences are applied at the following strength relative to the maximum available gradient of approximately 50 G/cm: $G_1 = (79\%, -47.3\%)$, $G_2 = (-47.3\%, 79\%)$, $G_3 = (41.5\%, -41.5\%)$, $G_4 = 5.2\%$, $G_5 = 15.7\%$, $G_6 = 31\%$, $G_7 = 23\%$, $G_8 = -11\%$, $G_9 = -7\%$, $G_{10} = -5\%$, $G_{11} = 37\%$, $G_{12} = (20.9\%, -20.9\%)$, $G_{13} = -G_{12}$, $G_{14} = 83\%$, $G_{15} = G_{14}$, and $G_{16} = 19\%$. All gradients are applied for 1 ms with a subsequent $\delta_g = 200$ $\mu$s recovery delay, except for the gradients $G_{12}$, $G_{13}$, and $G_{15}$ which are applied at 750 $\mu$s, 750 $\mu$s and 500 $\mu$s, respectively. The $\{^1\text{H}\}^{13}\text{C}$-$J$-INEPT sequence has been applied with $^{13}\text{C}$ detection while decoupling protons using the WALTZ-16 decoupling sequence.



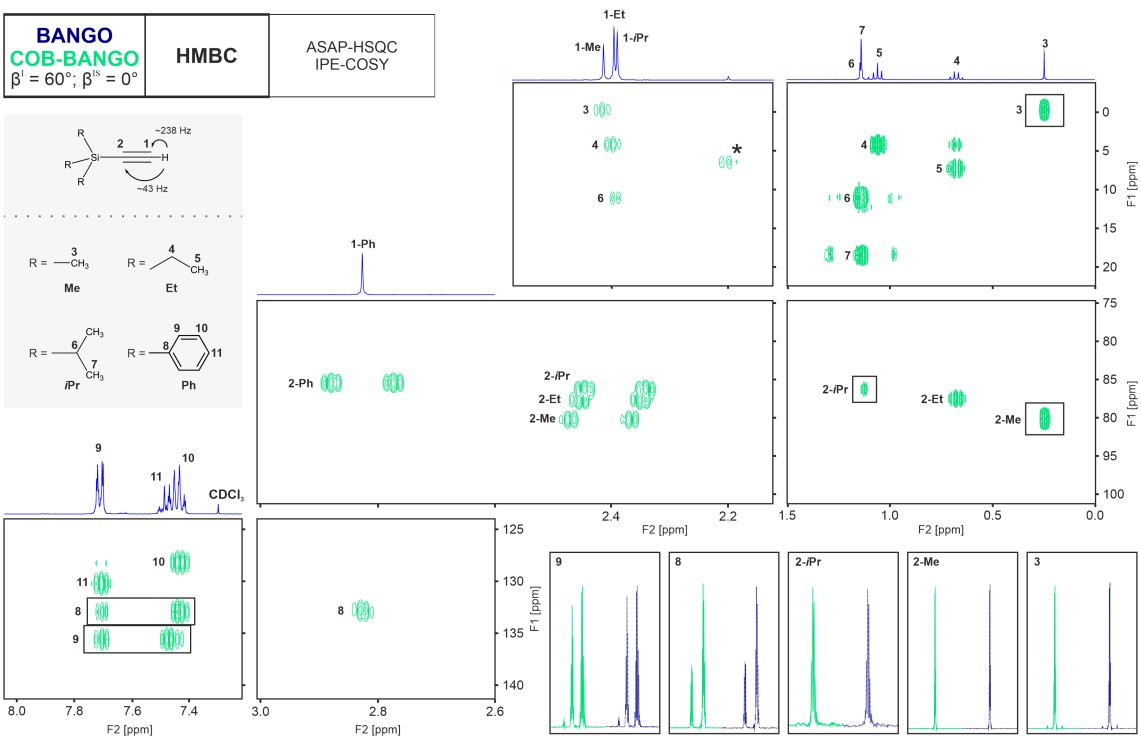

**Figure 5.** Experimental comparison of the HMBC sub-spectra extracted from the NORD supersequence when applying a BANGO or a COB-BANGO element for spin system selective excitation as described in the main text. In the top left corner, a schematic of the NORD sequence is shown, where the full pulse sequence is presented in Fig. 4. The BANGO and COB-BANGO are designed to apply a 60° excitation for isolated $I$ spins and store polarization along $z$ for protons of $IS$ spin systems for the subsequent ASAP-HSQC-IPE-COSY. The isotropic test sample consists of four silylic acetylenes as shown in the grey box on the left dissolved in CDCl$_3$. The relevant regions of the HMBC applied with a COB-BANGO are zoomed in the six boxes and are aligned with respect to each other and with corresponding 1D-$^1$H regions on top. Corresponding carbon and proton assignments are given within the 2D regions and in the 1D-$^1$H spectrum, respectively. Slices of selected, boxed signals are given as representative examples on the bottom left with their respective $^{13}$C assignments. The green solid spectra and the dashed blue spectra correspond to HMBC slices of the COB-BANGO and BANGO versions, respectively.



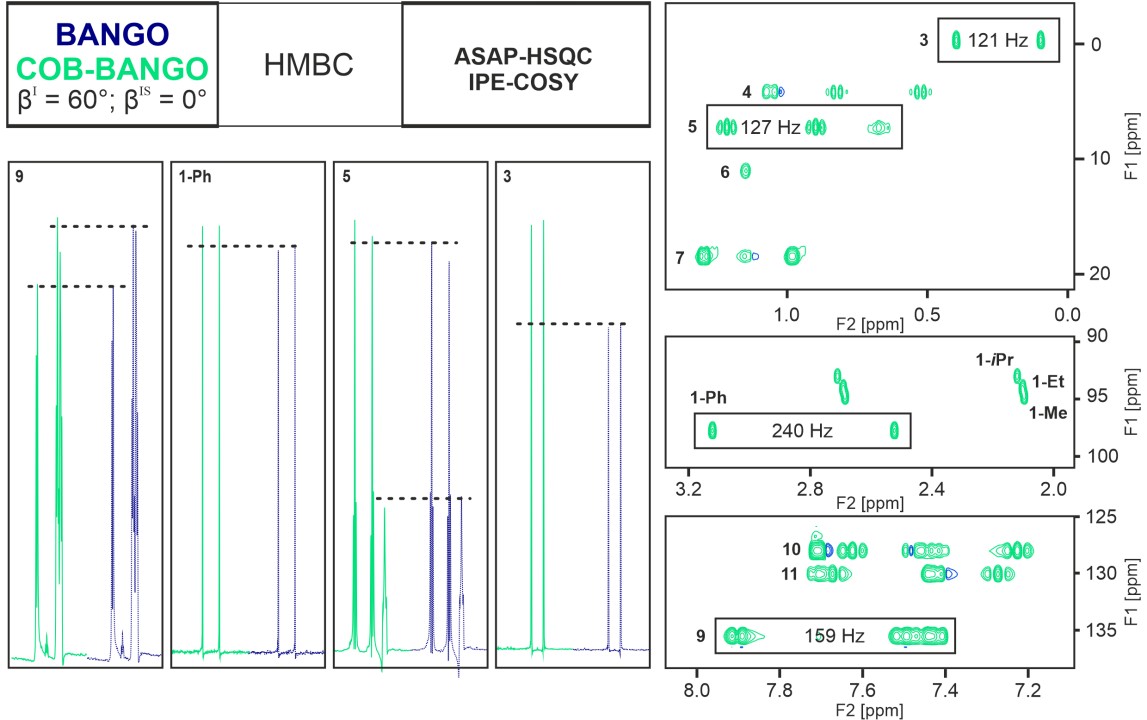

**Figure 6.** Experimental comparison of the ASAP-HSQC-IPE-COSY sub-spectra extracted from the NORD supersequence when applying a BANGO or a COB-BANGO element for storing polarization throughout the HMBC sequence. At the top left corner, a schematic of the NORD sequence is shown, where details of the sequence are given in Fig. 4. The BANGO and COB-BANGO are designed to apply a $60°$ excitation for the $I$ spins and storing polarization of the $IS$ spins for the subsequent ASAP-HSQC-IPE-COSY. On the right side 2D regions of the ASAP-HSQC-IPE-COSY are provided with the carbon assignments as introduced in Fig.5. Slices of selected, boxed signals are given on the left with their respective $^{13}$C assignment, whereas their respective $^1J_{CH}$ couplings are annotated within the boxes.

the 120-250 Hz range. We therefore used the silylic acetylene compounds from the previous demonstration experiments and dissolved them in the lytropic liquid crystal poly-$\gamma$-benzyl-L-glutamate (PBLG) with CDCl$_3$ and measured $^1$H,$^{13}$C one-bond couplings with our recently introduced COB-$J$-resolved-INEPT experiment (Woordes et al., 2025). The pulse sequence with experimental details is given in Fig. 4 B, where various BIRD-type elements derived from the $\pi$ rotations of Fig. 4 D-H were used for homonuclear decoupling in the indirect dimension. Resulting cross peaks for the one- and two-bond heteronuclear correlations involving triple bonds are summarized in Fig. 7.

The conventional BIRD with delays set for 145 Hz works well for small couplings, and, by accident, the largest coupling with $^1T_{CH} = 434\,\mathrm{Hz}$ ($\approx 3\cdot145$ Hz, three times the nominal coupling constant of the BIRD element). For couplings in the range of 284-334 Hz, no cross peak is observed. In contrast, all COB-type BIRD elements result in cross peaks for all carbons. It should be noticed that all COB-type elements were scaled as described in the figure caption to provide best intensities for the

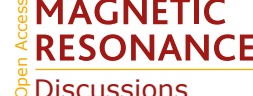

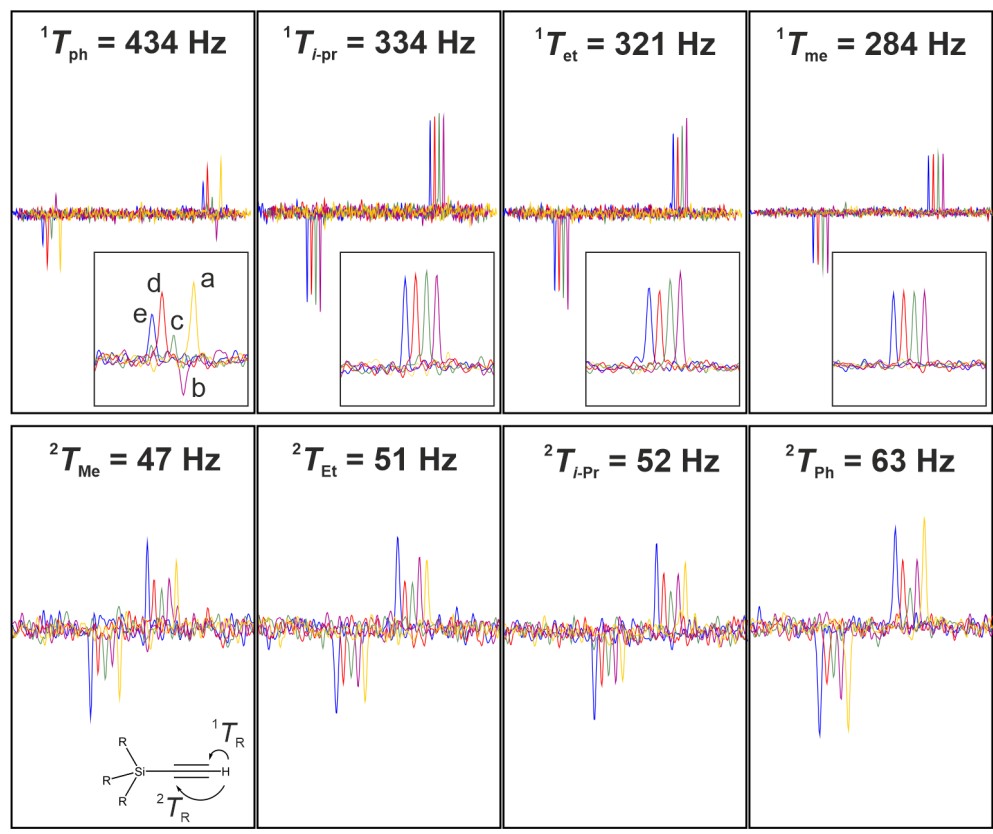

**Figure 7.** Comparison of acetylene signals for the various bilinear $\pi$-rotation elements using a $J$-resolved, homonuclear decoupled, $^{13}$C detected INEPT sequence. Shown are traces along the indirect $J$-dimension with heteronuclear coupling evolution and $^1$H,$^1$H homonuclear decoupling. Details of the pulse sequence are summarized in Fig. 4 B. For the measurement, the same silylic acetylene compounds as shown in Fig. 5 are dissolved in an 11% lyotropic mesophase consisting of PBLG/CDCl$_3$. Under partial alignment total $^1T_{CH}$ and $^2T_{CH}$ couplings from scalar and residual dipolar contributions range from 47 Hz to 434 Hz. The assignment and coupling size of each acetylene cross peak are indicated at the bottom left with the rest group (see gray box in Fig. 5). For the comparison, the sequence was tested using (a) BIRD, (b) COB-BIRD, (c) COB-BIRDcp, (d) COB3-BIRDcp-nd$_3$, and (e) COB3-BIRDcp-nd$_4$ bilinear rotations for homodecoupling. The BIRD filter was matched to 145 Hz transfer delays – which happens to result in a perfect transfer for 145 Hz and 3 * 145 Hz = 435 Hz. For covering all couplings with the best possible intensities, the delays for the COB-BIRD, COB-BIRDcp, COB3-BIRDcp-nd$_3$ and COB3-BIRDcp-nd$_4$ are scaled relative to the values given in the caption of Fig. 4 by 0.75, 0.75, 1.25, and 1.25, respectively.





entire 47-434 Hz range. For the largest coupling as well as for all small couplings differences can be seen with the COB3-type
elements performing best.

## 7 Discussion

A thorough analysis of various bilinear rotation elements resulted in two design principles that allow to produce all kinds of
compensated bilinear rotations from an existing compensated bilinear rotation or even an existing compensated INEPT-type
transfer element. The first principle is based on the finding that a common central $\pi$-rotation element is the essence of all
basic types of bilinear rotations. By selectively optimizing this central element, this allows us to equally improve all bilinear
rotations. In NMR spectroscopy, such robust $\pi$-rotation elements can be taken from existing ones, like the one derived for the
original COB-BIRD (Woordes et al., 2025), or e.g. the CAGE-BIRD sequence for compensation of homonuclear antiphase
evolution (Haller, 2021). The second construction principle is an extension of a previously derived one for universal rotation
pulses with flip angle $2\beta$ that are combined from two point-to-point pulses with effective flip angles $\beta$ (Luy et al., 2005). As
we could show, this construction principle can be translated from offset dependent single spin pulse shapes to $J$-dependent
transfer elements like INEPT as long as the elements are reduced to pulse only on one of the coupled spins. Instead of using
such a construction element, of course, a central $\pi$-rotation element can also be optimized from scratch, for which we refer the
interested reader to (Woordes et al., 2025).

All sequences introduced here have been applied with previously published shaped pulses, which have all been optimized for
[1]H,[13]C applications on a 600 MHz spectrometer, corresponding to a high-end system for typical small molecule applications.
If offset ranges different from 10 kHz on [1]H and 37.5 kHz on [13]C are desired, just shaped pulses may be re-optimized using
standard optimal control procedures, for which several optimization programs are readily available (de Fouquieres et al.,
2011; Goodwin and Vinding, 2023; Hogben et al., 2011; Goodwin et al., 2020; Maximov et al., 2008; Tošner et al., 2009;
Buchanan et al., 2024; Ehni and Luy, 2013; Kobzar et al., 2004, 2008; Koos et al., 2015; Lingel et al., 2020). Be aware that
the optimization of fully $J$-compensated pulse sandwiches like BUBI and BUBU (Ehni and Luy, 2013; Ehni et al., 2022) may
be quite demanding. If simple universal rotation pulses are applied simultaneously on the two heterochannels instead, a certain
compromise with respect to unspecified $J$-coupling evolution during the pulses must be expected. Pulse shapes may also be
re-optimized if higher compensation with respect to $B_1$ inhomogeneity is needed.

If the sequences shall be transferred to other spins, for example to [1]H,[15]N, the $J$-compensation can be scaled down to
smaller $J$-couplings, by correspondingly extending the delays in the central part of the bilinear rotations. The COB-BIRD, for
example, will work for a coupling range of 60-125 Hz (instead of 120-250 Hz), if all delays are doubled.

In actual applications, the COB-type bilinear rotations work very well for a large variety of heteronucler $J$ and dipolar cou-
plings. However, homonuclear couplings are not taken into account in the optimization process. Small ones are easily tolerated,
but if large and many [1]H,[1]H couplings are present, significant magnetization might be lost during the elements. Bilinear rota-
tions generally only work well as long as the condition $^1J_{CH} \gg \sum_i J_{HH,i}$ is sufficiently fulfilled. This is particularly an issue
with CH$_2$ groups with their large $^2J_{HH}$ couplings and the potentially high number of additional couplings. Since COB-type





sequences are significantly longer than the conventional doubly refocused delay, the loss of magnetization due to homonuclear couplings is also expected to be higher and easily explains the reduced gain observed for the CH$_2$ group compared to the methyl group in the ASAP-HSQC-IPE-COSY of Fig. 6. Equally, fast relaxing molecules may experience a larger loss in sensitivity for

the longer sequences. For a fair judgement of the different sequences, the duration should not be used directly to estimate the coupling evolution and relaxation effects. Due to odd pulse flip angles part of the magnetization is also stored along $z$ during the COB-type elements, reducing the strengths of the unwanted effects.

Regarding fully isotope labeled samples, robust bilinear rotations based on the BASEREX approach (Haller et al., 2019; Bodor et al., 2020; Sebák et al., 2022) may be obtained for a specific bandwidth if all carbon broadband pulses are replaced by

corresponding shaped pulses with controlled coupling evolution like the previously used REBURP pulse shape (Haller et al., 2019; Geen and Freeman, 1991).

As compensated COB-type BIG-BIRD sequences are generally possible, also corresponding compensated TIG-BIRD sequences (Briand and Sørensen, 1998) can be constructed. Spin state selectivity must be obtained with an additional transfer element, which still will have the restrictions of the conventional refocused delay approach. Only for special cases, for example

if pure antiphase magnetization for the $IS$ spin system is targetted, equally compensated sequences like the COB-INEPT (Ehni and Luy, 2012) are available for the extension.

## 8    Conclusions

Highly compensated bilinear rotations of the BIRD, TANGO, BANGO, and BIG-BIRD type have been introduced with $J$-compensation up to a $J$-coupling range of 120-750 Hz and offset ranges for $^1$H and $^{13}$C of 10 kHz and 37.5 kHz, respectively.

This has been achieved using a general principle, in which the central refocused delay common to all basic bilinear rotations is being optimized or constructed from existing elements. Particularly the latter has been achieved by using a construction principle originally derived for shaped pulse design for single spins. Effective ways of producing bilinear rotations with other demands on $J$-coupling ranges or offset/B$_1$ compensations are discussed.

We foresee direct applications for the different bilinear rotations introduced here in the area of partially aligned samples,

where a large range of $(J + D)$-couplings is unavoidable, or in NMR-service applications that also consider sp-hybridized carbons with typical $^1J_{\text{CH}}$ coupling constants around 250 Hz. Another obvious field will be $^{19}$F-based correlation experiments, for which one-bond couplings show a relatively wide distribution, and many more potential applications are thinkable.

*Data availability.* Spectra in JCAMP-DX and Bruker format together with Bruker pulse programs used for acquisition of example NMR spectra are available at DOI (DOI identifier will be added after revisionXXX)



*Author contributions.* Y.T.W. did all simulations and experiments and was involved in drawing/writing part of the manuscript. The initial idea, supervision and partly writing the manuscrpt was the responsibility of B.L..

*Competing interests.* There are no compeeting interests.

*Acknowledgements.* The authors are grateful for funding by the Deutsche Forschungsgemeinschaft (SFB 1527 HyPERION, project C01) and by the HGF-programme Information (43.35.02).



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
