# Peer review of "Robust Bilinear Rotations II"

_Magnetic Resonance, 2025_

## Author Response (AR1)

The authors very much thank the referees for their time and patience to go through the text and give us valuable advice. Please find our answers to the specific points raised in the following sections.

**Answer to Comments of Anonymous Referee #1**

**General comments**

In NMR spectroscopy, which allows to reveal the atomic resolution information about the chemical structure and molecular dynamics of molecules, the ability to manipulate spin coherences according to the ones desire is of great importance. Pulse sequence elements permitting rotations of chosen spin coherences, while leaving other ones unaffected are therefore highly needed. A class of such pulse sequence elements selectively and concurrently rotating the J-coupled IS spin coherences and non-coupled I spin coherences is usually termed Bilinear Rotation (BR). These elements usually come with inherent limitations in the correct-phase signal-maximized coherence transfer as the actual J-couplings of the system of interest may differ from the one for which the experiment has been designed. Additionally, the effects of hard pulse imperfections and magnetic field inhomogeneity come into play, imposing further bounds on the coherence transfer efficiencies. In this article, these issues have been successfully addressed using the new methods coined "COB", which stands for Coupling, Offset and $B_1$-inhomogeneity compensated pulse sequence elements. The use of pulse-delay calibrated spin evolution allows compensation with respect to the actual J-coupling modulation, while the application of optimal control derived offset and $B_1$-inhomogeneity compensated uniform rotation or point-to-point pulse shapes, allows to make these pulse sequence elements as robust as physically possible. According to the authors, these elements may tolerate an astounding variation in J-coupling of approximately 100-750 Hz, with chemical shift offset in the range of 10 kHz (37.5 kHz) in $^1$H ($^{13}$C) dimension.

>>> The authors agree very much with the general comment of the referee. A small detail: J-couplings are tolerated in the range of approximately 120-750 Hz, although also 100 Hz couplings are feasible in applications with accordingly reduced signal intensities.

**Specific comments**

In this article, these COB pulse sequence elements have been successfully applied to BR elements such as BIRD, TANGO, BANGO and BIG-BIRD, allowing for a highly robust distinction of J-coupled IS coherences and non-coupled I coherences, permitting on-demand concurrent manipulation of those two types of coherences. These BR elements, in general, come in handy when trying to design no-relaxation delay pulse sequences where evolution of chosen coherences is allowed, whilst suppressed for the other ones, in order to use the latter in the subsequent part of the experiment.

Also, BR elements are of prime importance in the fast and efficient real-time homonuclear decoupling elements like BASEREX. In the latter, the large typically one-bond heteronuclear coupling is used to distinguish J-coupled spin coherences IS from IX, where X is a different spin of the same nucleus as I, distant to heteronuclear spin S (with much smaller J-coupling between the spins S and X than between the spins S and I). In principle, the use of such elements up until now has been limited to low J-coupling range and chemical shift offsets. According to the authors, the use of COB principles allows to efficiently extend those

experiments for the cases of high (J-coupling or offset) heterogeneity which may be intrinsic to the sample. Such heterogeneity may be found, for instance, in the $^{19}$F-$^{1}$H coupled systems, which are gradually becoming of high interest in nowadays applications like drug chemistry, or even in biomolecular *in vitro* and *in vivo* studies. The NMR methods for these systems are slowly emerging, but until recently, required many experiments to cover the extent of $^{19}$F chemical shift range or $^{19}$F-$^{1}$H coupling values. Even though an example of such application would be very welcome, it is hard to show all of the applications of the method in one article and may easily go beyond its scope.

>>> We very much agree with the referee that $^{19}$F,$^{1}$H or $^{19}$F,$^{13}$C applications in principle would be very nice examples for COB-type transfer elements. Particularly $^{19}$F,$^{13}$C correlations over widely ranging one-bond couplings are interesting. However, for BR elements, the often very large $^{19}$F,$^{19}$F and $^{19}$F,$^{1}$H couplings will cause tremendous artefacts and need considerable thoughts for this type of applications. For $^{19}$F,$^{1}$H-correlations, on the other hand, both nuclei are at high natural abundance, preventing the use of BRs in classical applications like homodecoupling. As we believe that further development will be needed for broadly satisfying correlation experiments, we decided to not include such applications in the current manuscript.

The second example of the use of COB-BR elements may be, as has been shown by the authors in the article, in the studies of conformational dynamics of molecules and macromolecules using Residual Dipolar Couplings. In the latter, the use of an alignment medium to partially reduce the rotational averaging of the dipolar coupling interactions leads to efficient modulation of the so-called total (T) coupling (J-coupling plus Dipolar coupling). As a result, the apparent J-couplings will be largely modified positively or negatively to the extent imposed by the degree of alignment of the sample with typical variation of up to 30-50% of the nominal J-value. This modulation will occur differently for different parts of a molecule, encoding precious information on orientational and conformational averaging of the molecule. Normally, this phenomenon would modify the magnetization transfer efficiencies and if sufficient care is not taken, sometimes even phases of the resultant signals differently for different resonances of the same molecule in experiments based on INEPT or BR elements. This, would in turn require acquisition of many experiments with delays in the INEPT/BR elements designed to maximize the signals of every spin system in the aligned sample. The latter is not only difficult, but also time consuming, as the molecular diversity usually requires a few of such experiments with delays optimized to coupling values in the range of e.g. 90 to 200 Hz (for $^{13}$C$^{1}$H spin system). The latter arises due to the fact that the knowledge of the T-coupling values is targeted and usually only its approximate range may be estimated *a priori*. The use of COB principles coupled with INEPT and BR elements allows one to efficiently remove the dependence on the coupling and offset, to record all of these responses in one single experiment.

>>> The authors fully agree with everything said!

The full understanding of the article requires knowledge of the previous work of the authors to fully appreciate the technical details. It is, however, beyond the scope of the article to fully discuss the functioning of all of the BR elements and COB principles of design, while proper citation is correctly annotated by the references.

>>>Two major concepts have been introduced in the manuscript and we had to keep explanations within limits and therefore had to refer to previous work a lot. We apologize, but a proper derivation of all details necessary would be a project on its own. But indeed, we are working on a text which might become a review on bilinear rotations including recent developments, which may just be the text needed.

**Technical comments**

In the Figure 1., the discrimination between Uniform Rotation (UR) and Point-to-Point (PP) pulse shapes as implemented in the Figure 1C,E,G is available only at the right-hand side of the figure 1I, when discrimination of those pulse shape characteristics is needed already earlier to fully understand the Figure 1C. I would like to propose to shift the "legend" of UR and PP pulses to the top of the Figure 1, next to Figure 1A, which then may be made consistently 20% smaller to accommodate space for the mentioned change. In the same time, the Figures 1H and 1I could be made slightly bigger, as for the moment it is difficult to read the J-values and offsets in those figures. I am guessing that the "PP" placed not far away from "Phi" in the Figure 1H and 1I was supposed to be a subscript, but at the moment it is a bit too far from it and seems to be without any meaning. Also, it is difficult to distinguish UR 90° pulse shape from the hard 90° pulse shape in Figure 1C,E,G. One could improve this distinction by playing with the shade of grey or other visibly different color of choice (e.g. red) for the UR 90° pulse.

>>> We very much thank the referee for the detailed suggestions of changes! We reduced 1A and placed the legend for UR and PP pulses next to it. We also changed the color of the flanking pulses to red to distinguish them from the central sequence. The Figure caption has been extended accordingly.

In Figure 2F, there seems to be an excessive amount of tiny black spacers: beneath the second part of the rotation sandwich (x, x, -x, x, y) and beneath the second part of IS rotation sandwich (Delta_4, Delta_3, Delta_2). It also seems that these spacers are lacking in the figures of magnetization profiles as a function of J-value just beneath the Figure 2F. This figure should be revised accordingly. Also, from an aesthetic point of view, it would be useful to decrease the size of the pulse sequence scheme elements in Figure 2C,D,E,F and align their centers with the centers of the corresponding magnetization transfer profiles as a function of J placed just underneath.

>>> The referee has very good eyes! Indeed, an accident must have happened when preparing the figure that escaped our attention. We applied all advised actions.

In line 128, "In all cases but the BIG-BIRD profiles we overlaid simulated J-profiles as solid red lines over the experimental data for inner (uncoupled) and outer (coupled) multiplet components.", the simulated profiles are referred to red lines, while the green lines are shown in the figure. It would be helpful to make it consistently "red" in the plot and the Figure 3 caption, or change the text in the line 128 to "green".

>>>The figures originally were thought to contain the simulated J-profiles in red. However, after reading guidelines regarding color-blindness and b/w printing behavior, we changed the color to green. Apparently, we forgot to replace the in-text referencing, which we have changed now.

In lines 31, 39, 156 and 157, a question mark is found instead of a citation. Please, correct it.

>>>Ooops, we very much apologize. We identified the reasons and corrected the text accordingly.

In line 163, one may find "would be good to be replaceed" where it should read "would be good to be replaced", please correct.

>>> Thank you for pointing out the typo!

In line 190, "lytropic liquid crystal" should read "lyotropic liquid crystal", please correct.

>>> We thank the referee again and for the very positive review in general!

**Answer to Comments by Tom Barbara**

The applications for BIRD are extensive and the introduction of Chirp pulses is a significant improvement in the method. Some readers may be interested in another area that was worked on back in the late 80's early 90's and that is the application to deuterium NMR in solids and liquid crystals. One can broaden the bandwidth of double quantum coherence and quadrupolar order excitation. This also came out of the Pines lab with a modest contribution by myself: Barbara, Tycko and and Weitekamp J. Magn. Reson. 62,54 (1985) and other papers by Steve Wimperis.

>>>The analogy of composite pulses and pulse sequence elements for many different applications has indeed been used massively in the mid- and late 1980's and is related to the work presented here. Luckily, we can use optimal control based methods to fully screen physical limits and judge if a given composite pulse is already the best possible or not, which usually results in (slightly) improved performance. In addition, we can optimize highly compensated shaped pulses with arbitrary flip angles in a matter of hours, which further improves resulting sequences significantly. The relations derived mainly by Steve Wimperis and Malcolm Levitt may still be used today with correspondingly optimized shaped pulses to obtain well-performing pulse sequence elements. We gave a comparison with the original JC-BIRD published in the original BIRD paper by Garbow et al. in Reference (Woordes et al. 2025), where unfortunately the compensated element does not allow its use with antiphase magnetization. For many other previously derived composite pulses and corresponding transfer elements from the 1980's – we fully agree with the referee – a direct translation with the use of properly optimized shaped broadband pulses will allow remarkably robust transfer elements. A work still to be done!

**Answer to Comments by Malcolm Levitt**

This article describes continued methodological advances in the topic of bilinear rotations, which can induce different transformations for spins which are isolated and spins which have are coupled to others. Such manipulations may be used for a variety of useful purposes in solution NMR. The authors have previously introduced methods which make bilinear rotations more robust with respect to a variety of relevant parameter variations, such as

coupling variations. In this article the authors describe a construction principle for compensated bilinear rotations and also introduce an elaborate pulse sequence which exploits this methodology. The method is also applied to spin systems in partially oriented liquids.

The methodology described in this article is elaborate and I have to confess that the paper is not an easy read. The article assumes much familiarity with the arcane nomenclature for a large tranche of specialised pulse sequences. Nevertheless it is clear that the methodology is sound and the experimental results are impressive. There is little doubt that this is a valuable contribution to the NMR methodological literature, and that the described sequences significantly enhance the range and power of bilinear rotations.

>>>We certainly agree that today's multitude of specialized pulse sequences may feel like an impenetrable thicket. On the other hand, it may improve accuracy in the technical language to the ones familiar to it. We believe, a comprehensive review in the field is missing. As mentioned above, we are currently working on a text for bilinear rotations that should fulfill some of the need.

I note that a few citations appear to be broken. The terms "homomorph" and "homomorphous" should be defined within the relevant context.

>>> We fixed all broken citations, for which we would like to apologize. Homomorphous is used in the sense that the structure of the applied symmetry element is conserved. Be aware that this may hold only from pulses to pulse sequence element, as pulses other than inversions on the S spin are not allowed. But under the condition that spin states of the S spin are not changed, the relation of single spin rotations and two-spin J-evolution is even isomorphous We added an explanation to the text.

We believe that we could address all concerns raised adequately with the changes applied to the manuscript and we hope that its revised version will now be considered for full publication in Magnetic Resonance.

Sincerely yours,

Yannik T. Woordes and Burkhard Luy